# The Impact of the Mediterranean Diet and Lifestyle Intervention on Lipoprotein Subclass Profiles among Metabolic Syndrome Patients: Findings of a Randomized Controlled Trial

**DOI:** 10.3390/ijms25021338

**Published:** 2024-01-22

**Authors:** Beatriz Candás-Estébanez, Bárbara Fernández-Cidón, Emili Corbella, Cristian Tebé, Marta Fanlo-Maresma, Virginia Esteve-Luque, Jordi Salas-Salvadó, Montserrat Fitó, Antoni Riera-Mestre, Emilio Ros, Xavier Pintó

**Affiliations:** 1Clinical Laboratory, Bellvitge University Hospital, 08907 L’Hospitalet de Llobregat, Spain; bcandas@scias.com (B.C.-E.); barbara.fernandez@bellvitgehospital.cat (B.F.-C.); 2Clinical Laboratory, SCIAS-Barcelona Hospital, 08036 Barcelona, Spain; 3Department of Biochemistry, Molecular Biology and Biomedicine, Autonomous University of Barcelona (UAB), 08193 Barcelona, Spain; 4Lipids and Vascular Risk Unit, Internal Medicine Department, Bellvitge University Hospital, 08907 L’Hospitalet de Llobregat, Spain; emilic@bellvitgehospital.cat (E.C.); mfanlo@bellvitgehospital.cat (M.F.-M.); vesteve@bellvitgehospital.cat (V.E.-L.); ariera@bellvitgehospital.cat (A.R.-M.); 5Bellvitge Biomedical Research Institute (IDIBELL), 08907 L’Hospitalet de Llobregat, Spain; ctebe@idibell.cat; 6Center for Biomedical Research in Obesity and Nutrition Physiopathology Network (CIBEROBN), Carlos III Health Institute, 28029 Madrid, Spain; jordi.salas@urv.cat (J.S.-S.); mfito@researchmar.net (M.F.); eros@clinic.cat (E.R.); 7Nutrition Unit, Department of Biochemistry and Biotechnology, Rovira i Virgili University Human, 43204 Reus, Spain; 8Pere Virgili Health Research Institute (IISPV), San Joan de Reus University Hospital, 43204 Reus, Spain; 9Hospital del Mar Medical Research Institute (IMIM), 08003 Barcelona, Spain; 10Department of Medicine, School of Medicine, Bellvitge Campus, Barcelona University, 08007 Barcelona, Spain; 11Lipid Clinic, Department of Endocrinology and Nutrition, Institut d’Investigacions Biomèdiques August Pi Sunyer (IDIBAPS), Hospital Clínic, 08036 Barcelona, Spain

**Keywords:** metabolic syndrome, Mediterranean diet, lipoprotein precipitation, nuclear magnetic resonance lipid profile, advanced lipoprotein tests, small dense LDL

## Abstract

Metabolic syndrome (MetS) is associated with alterations of lipoprotein structure and function that can be characterized with advanced lipoprotein testing (ADLT). The effect of the Mediterranean diet (MedDiet) and weight loss on the lipoprotein subclass profile has been scarcely studied. Within the PREDIMED-Plus randomized controlled trial, a sub-study conducted at Bellvitge Hospital recruiting center evaluated the effects of a weight loss program based on an energy-reduced MedDiet (er-MedDiet) and physical activity (PA) promotion (intervention group) compared with energy-unrestricted MedDiet recommendations (control group) on ADLT-assessed lipoprotein subclasses. 202 patients with MetS (*n* = 107, intervention; *n* = 95, control) were included. Lipid profiles were determined, and ADLT was performed at baseline, 6, and 12 months. Linear mixed models were used to assess the effects of intervention on lipoprotein profiles. Compared to the control diet, at 12 months, the er-MedDiet+PA resulted in a significant additional 4.2 kg of body weight loss, a decrease in body mass index by 1.4 kg/m^2^, reduction in waist circumference by 2.2 cm, decreased triglycerides, LDL-cholesterol and non-HDL-cholesterol, and increased HDL-cholesterol. In er-MedDiet+PA participants, ADLT revealed a decrease in small dense-LDL-cholesterol (sd-LDL-C), intermediate-density lipoproteins, VLDL-triglyceride, and HDL-Triglyceride, and an increase in large LDL and large VLDL particles. In conclusion, compared to an ad libitum MedDiet (control group), er-MedDiet+PA decreased plasma triglycerides and the triglyceride content in HDL and VLDL particles, decreased sd-LDL-C, and increased large LDL particles, indicating beneficial changes against cardiovascular disease.

## 1. Introduction

Cardiovascular disease (CVD) remains the leading cause of morbidity and mortality worldwide [1,2]. The concentration of low-density lipoprotein cholesterol (LDL-C) is tightly linked to CVD mortality [3,4] and is the main target of CVD prevention strategies [5]. However, ischemic events also occur in individuals with an LDL-C concentration below the cut-off value used to define increased cardiovascular risk [6], particularly in patients harboring metabolic syndrome (MetS) [7]. MetS is a clinical condition with insulin resistance and central obesity leading to glucose intolerance, dyslipidemia, and increased blood pressure as key components. Managing MetS requires modifying lifestyle habits, such as reducing weight through dieting and increasing physical activity [8]. Higher adherence to the Mediterranean diet (MedDiet) has a beneficial impact on lipid alterations and other components of the MetS [9,10] and is also associated with reduced mortality related to this disorder [11,12,13]. The PREDIMED trial has shown that the MedDiet has a protective effect against CVD [14,15]. This salutary effect can be attributed to the myriad beneficial nutrients and bioactives contained in MedDiet foods, such as vegetable proteins, monounsaturated and polyunsaturated fatty acids, dietary fiber, vitamins, non-sodium minerals, and polyphenols [16] that are contained in extra-virgin olive oil, whole grains, nuts and a wide variety of fruits and vegetables that are characteristic foods of the MedDiet [17]. In this sense, it has been reported that the consumption of polyphenol-rich extra-virgin olive oil decreases the atherogenicity of LDL particles [18].

Each type of lipoprotein, including LDL, can be sorted into subclasses according to differing size, density, and composition. Because of distinct physical properties, the impact of each lipoprotein subclass on cardiovascular risk is also different [19]. MetS patients have abnormal lipid profiles consisting of a “lipid triad” of (1) increased triglycerides (TGs), (2) decreased high-density lipoprotein cholesterol (HDL-C), and (3) small dense LDL (sd-LDL) particles as the dominant subclass of LDL [20]. sd-LDL particles are more atherogenic, in part because a higher number of sd-LDL particles than large LDL (lLDL) particles is needed to carry the same amount of cholesterol, and the higher the number of LDL particles, the higher the risk of CVD [21]. sd-LDL particles also promote atherosclerotic plaque rupture, which triggers ischemic events [22].

In recent years, new methods that characterize lipoproteins more accurately have been developed, including advanced lipoprotein testing (ADLT) based on Nuclear Magnetic Resonance Spectroscopy (NMR) [23] and novel precipitation assays, to directly quantify sd-LDL-C. However, the effects of the MedDiet on the physicochemical properties of lipoproteins have been scarcely studied. In a small study, polyphenols from olive oil and from thyme were associated with an improvement in lipoprotein particle atherogenic ratios and profile distribution of lipoprotein subclasses [17]. In addition, in a sub-study of the PREDIMED trial in which lipoproteins were profiled by NMR [24], the MedDiet enriched with nuts shifted lipoprotein subfractions to a less atherogenic pattern.

The aim of this study was to evaluate the effect of an intensive weight loss program based on an energy-reduced traditional MedDiet (er-MedDiet), physical activity (PA) promotion, and behavioral support (er-MedDiet+PA) on the physicochemical properties of lipoproteins assessed by ADLT, in comparison with an energy-unrestricted MedDiet (control group) after 6 months and 1-year of follow-up.

## 2. Results

The baseline clinical characteristics of study subjects are depicted in Table 1. All subjects included in the study were obese or overweight. Of these, 58.4% were on lipid-lowering treatment. Conventional lipid profiles were obtained during baseline visits.

During the baseline visit, most participants were at conventional lipid profile goals (Table 1). The percentage of those with LDL-C ≤ 3.37 mmol/L, HDL-C ≥ 1.04 mmol/L, and non-HDL-C ≤ 4.14 mmol/L was higher in the intervention group. However, mean LDL-C and mean HDL-C were similar between groups, and mean non-HDL was higher in the control group (*p* = 0.045). Mean LDL-C in the intervention group was lower than the upper cut-off value defined by the National Cholesterol Education Program (NCEP) [25].

Baseline energy, nutrient intake, physical activity, and changes at 6 and 12 months by intervention group are displayed in Table 2. As shown, participants in the intervention group followed an energy-restricted diet, with a lower intake of carbohydrates and a higher intake of monounsaturated fat than the control group. Physical activity was significantly higher at 6 months in the intervention group and nearly significantly higher at 12 months.

Advanced lipoprotein profiles were also determined during baseline visits (see Appendix A). The proportion of LDL-C content in small LDL particles and the proportion of particle content in each lipoprotein subclass were calculated as percentages. The mean sd-LDL-C was higher than the upper cut-off value of the reference interval [26]. The predominant subclasses of LDL particles were sd-LDL, which accounted for 61.2% of the total LDL particle number (LDL-P), and small HDL particles (sHDL-P) were the dominant subclass of total HDL particle number (HDL-P), representing 66.7% (Appendix A). These results are consistent with the phenotype of MetS dyslipidemia.

### 2.1. Effect of MedDiet and er-MedDiet+PA Based Intervention on Anthropometric Characteristics and Lipid Profile

Estimated coefficients and *p*-values were used to quantify the estimated effect of the MedDiet and er-MedDiet+PA on changes in anthropometric characteristics and lipid profiles that occurred between the baseline and follow-up visits. In Table 3, the more relevant variables from mixed model analysis are presented. Results of the intervention at 6 and 12 months are adjusted according to group, time, interaction group and time, sex, age, the administration of lipid-lowering treatments, and smoking status.

#### 2.1.1. Anthropometric Variables

Participants in the MedDiet control group sustained an improvement in all anthropometric variables at 6 months and 1 year. Table 3 shows that at 6 months of follow-up, the variables BMI and waist circumference decreased significantly in the final adjusted model. Body weight decreased by 1.7 kg at 6 months and 1.7 kg at 12 months. A further weight loss of 3.9 kg (*p* < 0.01) and 3.9 kg (*p* < 0.01) and a reduction in waist circumference of 2.9 cm (*p* < 0.01) and 2.2 cm (*p* < 0.01) at 6 months and 12 months, respectively, was estimated in the er-MedDiet+PA intervention group in comparison with the control group (*p* < 0.01) (Figure 1). At 6 months, 4.2% of subjects in the control group and 38.5% of subjects in the intervention group achieved weight loss of at least 8%, and these percentages were 5.3% and 39.8%, respectively, at 12 months.

#### 2.1.2. Lipid Profile

The effect of traditional MedDiet on the lipid panel evaluated by NMR is displayed in the first two columns of Table 3. The control group had an increase in HDL-C and a decrease in serum TG concentrations. These results were significant at 6 months (*p* = 0.05 and *p* < 0.01, respectively). LDL-C and non-HDL-C concentrations decreased at 12 months by 0.26 mmol/L (*p* < 0.01) and 0.37 mmol/L (*p* < 0.01), respectively. The additional effect of er-MedDiet+PA on ADLT measurements is shown in the last two columns of Table 3. In comparison with the control group, the er-MedDiet+PA-based intervention program led to a significant further reduction in TGs (*p* = 0.021) (Appendix A) and to an increase of 0.1 mmol/L of HDL-C (*p* < 0.01) (Appendix A). Also, an increment in LDL-C and non-HDL-C concentration was detected in the intervention group (*p* = 0.05 and *p* < 0.01, respectively).

#### 2.1.3. Advanced Lipoprotein Tests

In the control group, sd-LDL-C and IDL-C decreased by 0.13 mmol/L (*p* = 0.05) and 2.72 mg/dL (*p* = 0.05) at 12 months, and HDL-TG and HDL-P decreased at 6 months (4.19 mg/dL, *p* = 0.04 and 2.67 µmol/L, *p* = 0.046, respectively). In comparison with the control group, in the intervention group, a further decrease in sd-LDL-C and sd-LDL-C/LDL-C (%) of 0.21 mmol/L and 9.23% was observed at 6 months (*p* = 0.016, *p* < 0.01, respectively, and also a decrease in very low-density lipoprotein-TG (VLDL-TG) and large VLDL particles (lVLDL-P) of 42.3 mg/dL (*p* = 0.039) and 0.73 nmol/L (*p* = 0.014), respectively at 12 months. No changes in LDL-P were observed. These results are shown in Table 3 and Figure 2. Effect on VLDL-C, VLDL-P, LDL, and HDL advanced lipid profile results are shown in Appendix A. Analyses of data in Table 3 were also performed separately by sex, and non-clinically relevant differences were observed (Appendix A).

## 3. Discussion

The results of the original PREDIMED trial demonstrated that a non-energy-restricted MedDiet reduced the incidence of CVD among participants at high cardiovascular risk who were mostly overweight/obese [14]. In PREDIMED, the energy-unrestricted MedDiet intervention improved the lipid profile and led to small reductions in waist circumference and body weight [27]. On the other hand, in the last decades, lipoprotein assays have been developed, which are able to detail the composition of lipoproteins and can distinguish between their subclasses [28]. These advanced methods can be used to update our understanding of the effects of diet and lifestyle on lipid metabolism.

We compared the effects of an energy-unrestricted traditional MedDiet and an er-MedDiet+PA intervention on body weight and lipid profiles, including ADLT, in MetS patients enrolled in the PREDIMED-plus trial at Hospital Universitari de Bellvitge. All patients were overweight or obese, and 73.8% were treated with lipid-lowering drugs (Table 1). At baseline, most participants had no overt dyslipidemia judging from their conventional lipid profiles (Table 1), and although their mean LDL-C was lower than the upper limit defined by the NCEP [25], the mean sd-LDL-C was higher than the upper limit of the reference interval [29]. In addition, their mean LDL-P number was higher than the treatment target value, as defined by the American Association of Clinical Endocrinologists [30]. These data illustrate that changes in the concentration of sd-LDL-C and LDL-P do not always go in parallel with changes in LDL-C. Furthermore, sd-LDL-C and LDL-P remained abnormally elevated even in patients classified as low risk by their LDL-C level. Importantly, the dominant subclasses of LDL and HDL particles were found to be sLDL-P and sHDL-P, respectively (Table 3), values that are consistent with the MetS phenotype. MetS subjects are thought to have a smaller mean HDL particle size, and it has been hypothesized that this alteration is linked to inflammation [31]; however, this concept needs to be supported by further research [32].

### 3.1. Anthropometric Variables

Adoption of the (energy-unrestricted) MedDiet led to a slight improvement in all anthropometric variables at 6 months and 1 year of intervention (Table 3). Although patients in the intervention group did not reduce total energy intake, they increased energy expenditure and improved diet quality, with a marked decrease in carbohydrate intake and an increase in monounsaturated fat consumption. These results are consistent with those observed in the PREDIMED trial. In comparison, the er-MedDiet+PA intervention had a noticeable impact on body weight and waist circumference at 6 and 12 months. As expected, the er-MedDiet+PA intervention program was more effective than the MedDiet without energy restriction in achieving the weight loss targets. It has been repeatedly shown that overweight or obese persons following an er-MedDiet with or without enhanced PA lose weight, and this holds as long as the diet has energy curtailed [33].

### 3.2. Conventional Lipid Profile

After 6 months of MedDiet, a decrease in TGs and an increase in HDL-C concentrations was observed in both study arms. After 12 months, these differences decreased in magnitude and lost statistical significance in the control group, while they remained significant in the intervention group. These findings are consistent with those reported in a recent meta-analysis of RCTs of MedDiet for MetS components [9]. Regarding the er-MedDiet+PA intervention, raised HDL-C and reduced TG concentrations were observed at 12 months compared with the conventional MedDiet, highlighting the importance of the intensive intervention on these two components of the lipid triad. These results suggest that MedDiet combined with traditional health care has a beneficial effect on LDL-C and non-HDL-C levels beyond what is accomplished with lipid-lowering treatment alone. Furthermore, the er-MedDiet+PA led to greater improvement in lipid variables linked to MetS, such as TG and HDL-C. Despite this improvement in TG and HDL metabolism, and despite the weight loss, an increment in LDL-C and non-HDL-C concentration was detected.

### 3.3. Advanced Lipid Profile

With respect to ADLT measurements, sd-LDL-C, IDL-C, and HDL-TG, but not LDL-P, significantly decreased in response to the traditional MedDiet. sLDL-P number did not change significantly, but a non-significant negative regression coefficient was observed for this variable at 12 months. In addition, the cholesterol content in sd-LDL decreased significantly at 12 months. It has been well established that caloric restriction and exercise have a favorable effect on TG and lipoprotein metabolism [34]. However, the effect of the traditional MedDiet on LDL-P concentration and subclass distribution has been scarcely studied. In previous studies with MedDiet that included subjects with MetS, increases in LDL size and a favorable redistribution of cholesterol among the different LDL particles were observed [24,35,36], whereas in a study of healthy subjects, the MedDiet had no effect [37]. Although in the current study, LDL-P did not change, the MedDiet was associated with an improvement in LDL particle composition and sd-LDL-C was reduced. In addition, HDL-TG was reduced at 6 months; as far as we know, this is the first study that has demonstrated this effect. Previous studies have reported higher HDL-TG in patients with carotid atherosclerosis in association with other biomarkers linked to the MetS [38]. A decrease in HDL-P was also observed. A reasonable explanation for these results is that a change in HDL composition had occurred. However, no significant changes in HDL size or HDL sub-classes were observed. HDL-P and sHDL-P are inversely related to CVD mortality and even all-cause mortality in patients with coronary artery diseases [39]. A nearly significant decrease in IDL-C and IDL-TG (major determinants of pro-atherogenic remnant lipoproteins) was also observed after the conventional MedDiet.

In comparison with the standard MedDiet, the er-MedDiet+PA-based intervention led to a significant reduction in sd-LDL-C and the sd-LDL-C/LDL-C ratio but no changes in LDL-P. In accordance with our results, in a recent nutritional RCT, no decrease in LDL-P concentration was found in individuals following a low-carbohydrate diet who had lost weight [40]. Expectedly, the er-MedDiet+PA intervention had a beneficial effect on variables related to TG metabolism at 12 months. VLDL-C was reduced with borderline statistical significance (*p* = 0.056), and a similar trend was observed for the VLDL-C+IDL-C concentration (*p* = 0.092). The er-MedDiet+PA intervention also induced a decrease in VLDL-TG, and, as a consequence, lVLDL-P was also reduced, and LDL particle size increased—changes that are recognized as anti-atherogenic [21]. HDL-T also tended to decrease with the er-MedDiet+PA intervention. It should be noted that a moderate decrease in carbohydrate consumption and a moderate increase in monounsaturated fat consumption was observed in the intervention group compared to the control group, and these changes could have influenced the lipid responses. Low- or very-low-carbohydrate diets (so-called ketogenic diets), which usually are reciprocally enriched in fat, are superior to low-fat diets in improving cardiometabolic risk due to a TG-lowering and HDL-C-raising effect, with only a negligible effect on LDL-C [41]. Also, there is ample clinical trial evidence that such carbohydrate-restricted dietary interventions increase LDL peak particle size and decrease the numbers of total and small LDL particles [42]. Thus, diets lower in carbohydrates and higher in fat, such as the er-MedDiet used in PREDIMED-Plus, have the potential to improve both the standard and the advanced lipid profile. High-fat, carbohydrate-restricted diets improve atherogenic dyslipidemia and insulin resistance [25,27,30,31,32] because carbohydrate consumption increases hepatic TG synthesis and induces insulin secretion that leads to inhibition of lipolysis, enhanced delivery of fatty acids for hepatic esterification and overproduction and secretion of large TG-rich VLDL particles. In addition, other potential mechanisms might be involved in the interaction between MedDiet and the effects of statins on lipid metabolism, i.e., the reduction in plasma insulin observed with a carbohydrate-restricted MedDiet may decrease the expression of the HMG-CoA reductase gene, with ensuing lower secretion of VLDL-C, which, in turn, would account for the reduction in LDL-C [43].

These large VLDL exchange TG for cholesteryl ester in both LDL and HDL. TG-rich LDL is a preferred substrate for hepatic lipase that favors the production of sd-LDL particles [44,45], while TG enrichment of HDL correlates inversely with the particles’ functionality [46] and is a good marker of cardiometabolic risk [38]. These changes in lipid metabolism are observed even in the absence of significant weight loss [47]. Aerobic exercise improves atherogenic dyslipidemia, among other mechanisms, by increasing lipoprotein lipase activity and decreasing fasting and post-prandial serum TG [48].

Although the er-MedDiet-PA intervention had an impact on lipid variables linked to TG, a similar effect was not observed for variables linked to cholesterol. While LDL-C and non-HDL-C levels decreased in the standard MedDiet group, they increased in the er-MedDiet+PA group. This suggests that the cholesterol-lowering effect was related to the MedDiet dietary regime alone, while adding energy restriction and exercise in the er-MedDiet+PA intervention counteracted this effect. In some studies, it has been observed that LDL-C increases after a session of moderate-intensity exercise [49] or when adhering to a low-calorie diet [50]. Whether energy restriction affects LDL-C levels is controversial. Thus, there is evidence in favor of modest LDL-C lowering in response to weight loss [51]. However, there are notable exceptions from seminal studies in which weight loss was unassociated with LDL-C changes [52,53], including preliminary evidence from the PREDIMED-Plus study [54]. Beyond weight loss, macronutrient changes in the diets (i.e., saturated fatty acids) may explain these discrepancies.

Our study has limitations. Participants in the PREDIMED-Plus trial are predominantly older white Spanish individuals with overweight or obesity harboring the MetS, which limits the generalizability of the results to other populations. Despite this, one strength of this study lies in the homogeneity of the cohort, which increases the internal validity of the findings by avoiding confounding factors related to socioeconomic status or educational level. We have also controlled for other possible confounders using multivariable statistical models. Another strength of our study is the longitudinal analysis carried out in a homogeneous and sizable cohort. Also, all participants were well characterized with clinical and laboratory variables related to MetS, including dietary components and physical activity. A further strength is the use of ADLT for lipid and lipoprotein analyses, allowing for a broad and comprehensive characterization of the spectrum of plasma lipoprotein species and their response to er-MedDiet+PA. Finally, the design of the statistical study allowed us to obtain reliable results since many possible confounders were considered.

## 4. Materials and Methods

### 4.1. Study Design

This is a prospective cohort study of participants from the PREDIMED-Plus trial. The PREDIMED-Plus study is an ongoing multicenter, parallel-group, randomized, single-blind clinical trial involving 6874 participants that were recruited in 23 Spanish centers. The aim of the PREDIMED-Plus study is to assess the long-term effects of an intensive weight loss program on cardiovascular events and mortality in comparison with a MedDiet (control group) (protocol available at https://www.predimedplus.com/wp-content/uploads/2018/11/Protocolo-PREDIMED-Plus_Eng.pdf; accessed on 9 June 2023). Participants were randomly assigned, in a 1:1 ratio, to one of two groups: an intensive weight-loss intervention group or a control group. In summary, the intensive weight loss program consisted of an er-MedDiet together with the promotion of physical activity and behavioral support for specific weight loss goals that included an average reduction in baseline body weight of over 8% and an average reduction in waist circumference of over 5% in the first six months and maintaining these reductions throughout the duration of the study. The er-MedDiet intervention targeted a reduction of approximately 30% in estimated energy requirements, which represented a reduction goal of approximately 600 kcal/day [29]. In addition, the er-MedDiet aimed to promote better overall diet quality through the limitation of certain foods such as sugar-sweetened beverages, butter and cream, red and processed meats, added sugars, sweets, pastries, and refined grains, including white bread, in favor of whole grains. Physical activity promotion included a face-to-face educational program [55] aimed at gradually increasing participants’ aerobic physical activity levels to meet at least the World Health Organization (WHO) guidelines based on age, the health status of the participants [56], and static exercises to improve endurance, strength, flexibility, and balance. Participants in the control group were encouraged to follow an unrestricted energy MedDiet, had biannual educational sessions on the traditional Medical Diet with ad libitum caloric intake, and received usual attention to general lifestyle recommendations [57].

The trial was registered in 2014 at the International Standard Randomized Controlled Trial registry as number 89898870 (http://www.isrctn.com/ISRCTN89898870; accessed on 9 June 2023). The primary outcome of this sub-study of the Predimed-Plus trial was to analyze the effect of an intensive weight loss program on the lipoprotein profile evaluated by ADLT.

### 4.2. Study Subjects

The participants were community-dwelling men aged 55–75 years and women aged 60–75 years without a documented history of CVD at baseline, with a body mass index ≥ 27 and <40 kg/m^2^, and with at least 3 of the 5 criteria for MetS [58]. Participants were recruited and randomly allocated, in a 1:1 ratio, to either the intervention or the control group. Only those participants who visited the Hospital Universitari de Bellvitge (L’Hospitalet de Llobregat) were included in this study. Intervention group participants were prescribed an er-MedDiet+PA and received personal behavioral support following the PREDIMED-Plus intervention protocol. Control group subjects were prescribed the original unrestricted-energy MedDiet and received conventional health care recommendations. The flow chart is shown in Appendix A.

Anthropometric measures were recorded and blood samples were collected. Overweight was defined as a BMI between 25 and 30 kg/m^2^, and obesity as a BMI ≥ 30 kg/m^2^. Dietary data were collected using a validated semiquantitative food frequency questionnaire, including 143 items commonly consumed in Spain [29]. The adherence to MedDiet was assessed by a 17-item questionnaire [59]. Physical activity was measured in MET*min/week using the Regicor Short Physical Activity Questionnaire [60].

### 4.3. Methods

#### 4.3.1. Conventional Lipid Profile

Blood samples were taken after 12 h of fasting during the baseline visit and then every six months for a year. The samples were collected in tubes that contained a separating gel but did not contain a coagulant (Vacuette ref: 456069). The tubes were centrifuged at 1500× *g* for 10 min (6K15 SIGMA centrifuge). The serum was immediately separated and stored at −80 °C until they were analyzed.

Total cholesterol was measured by molecular absorption spectrometry at 505 and 700 nm. By the action of cholesterol esterase, cholesterol esters are separated into free cholesterol and fatty acids. The enzyme cholesterol oxidase catalyzes the reaction that transforms free cholesterol to cholest-4-en-3-one and hydrogen peroxide. In the presence of peroxidase, phenol, and 4-aminophenazone, hydrogen peroxide forms a red quionimine dye. The chromatic intensity of the dye is directly proportional to the concentration of cholesterol in the sample [61,62].

The HDL-C concentration was measured by molecular absorption spectrometry. When exposed to a detergent, non-HDL lipoproteins, including chylomicrons, VLDL, and LDL, form a water-soluble complex wherein the enzymatic reactions of cholesterol esterase and cholesterol oxidase are inhibited so that only HDL particles can react with the two enzymes.

TGs were measured by molecular absorption spectrometry at 505 and 700 nm. This method uses lipoprotein lipase, glycerokinase, glycerol phosphate oxidase, and peroxidase. The lipoprotein lipase hydrolyzes the TGs to free fatty acids and glycerol; the latter is oxidized to dihydroxyacetonephosphate and hydrogen peroxide; subsequently, the peroxide reacts with 4-aminophenazone and 4-chlorophenol to give a red dye. The intensity of the dye is directly proportional to the concentration of TGs present in the sample.

All the analyses were performed at Cobas c501 (Roche^®^ Diagnostics Basel, Switzerland). The reagents used in the homogenous automatized assays were Cholesterol Gen 2 (Ref: 03039773190) for Cholesterol, HDL-Cholesterol plus 3rd generation for HDL-C (Ref: 05168805190), and TRIGL Triglycerides (Ref: 08058687190) for TG.

All analytical series were validated by measuring internal controls with known concentrations provided by Bio Rad Laboratories (Hercules, CA, USA). Currently, the laboratory participates in an external quality control program, Referenzinstitut für Bioanalytik (Bonn, Germany), to verify the accuracy of the results.

The percentage of participants that were at the National Cholesterol Education Program (NCEP) Adult Treatment Panel-III (ATP-III) lipid goals was evaluated [25]. These goals were LDL-C ≤ 3.37 mmol/L, HDL-C ≥ 1.04 mmol/L, TG ≤ 1.70 mmol/L, and non-HDL-C ≤ 4.14 mmol/L [25].

LDL-C was estimated using the Friedewald equation, and non-HDL-C was calculated by subtracting HDL-C from total cholesterol [63].

#### 4.3.2. Advanced Lipoprotein Precipitation Assays

sd-LDL-C concentrations were determined using a lipoprotein precipitation method that had been adapted to clinical routine laboratory settings [26,64]. The precipitation assay was carried out 2 weeks after blood extraction. To isolate sd-LDL particles, 300 uL of the sample was combined with 300 uL precipitation reagent (150 U/mL heparin-Na+, catalog #H3393; Sigma-Aldrich (Merck KGaA, Darmstadt, Germany); 90 mM MgCl_2_). The mixtures were then incubated at 37 °C for 10 min, placed at 0 °C for 15 min, and then centrifuged at 21,913× *g* (14,000 rpm) for 15 min at 4 °C (centrifuge catalog #6K15; Sigma-Aldrich). Lipoproteins whose density was <1.044 g/mL remained at the bottom of the tube, forming a yellow precipitate. The supernatant contained both HDL and sd-LDL particles, whose density fell between 1.044 and 1.063 g/mL. The concentration of supernatant-derived HDL-C and total cholesterol were determined using a Cobas 8000 modular analyzer (Roche^®^ Diagnostics) [65]. Since the supernatant only contained cholesterol from HDL and sd-LDL lipoproteins, the sd-LDL-C concentration was estimated by subtracting the HDL-C from the concentration of total cholesterol. The reference value for the concentration of sd-LDL-C is 0.04–0.47 mmol/L [26].

#### 4.3.3. Advanced Lipoprotein Profile by NMR Spectroscopy

As previously reported, 200 μL of serum was diluted with 50 µL deuterated water and 300 µL of 50 mM phosphate buffer solution (PBS) at pH 7.4. 1H-NMR spectra were recorded at 310 K on a Bruker Avance III 600 spectrometer (Bruker BioSciences, Madrid, Spain) operating at a proton frequency of 600.20 MHz (14.1 T) [28].

Complete lipoprotein profiles were determined using the Liposcale^®^ test. Liposcale tests are based on a 2D diffusion-ordered 1H NMR methodology to characterize lipoprotein subclasses, such as size, lipid composition, and number of particles [28]. These profiles included (1) TG and cholesterol concentrations; (2) the size and number of VLDL, LDL, and HDL particles; and (3) the number of large, medium, and small subclasses of VLDL, LDL, and HDL particles. The particle concentrations and size were derived from the NMR signals of the univocally associated methyl lipid groups that vary between the lipoprotein subclasses.

The reference values for the advanced lipid profile are as follows: VLDL-P (24.8–50.0) nmol/L; lVLDL-P (0.70–1.18) nmol/L; mVLDL-P (2.50–5.37) nmol/L; sVLDL-P (21.7–44.1) nmol/L; LDL-P (1128–1498) nmol/L; lLDL-P (163–214) nmol/L; mLDL-P (320–513) nmol/L; sLDL-P (598–786) nmol/L; HDL-P (25.2–33.1) µmol/L; lHDL-P (0.23–0.31) µmol/L; mHDL-P (7.98–11.0) µmol/L; sHDL-P (16.7–22.4) µmol/L; VLDL-C (4.66–13.8) mg/dL; LDL-C (111–149) mg/dL; HDL-C (44.4–66.7) mg/dL; VLDL-TG (36.4–71.4) mg/dL; LDL-TG (12.6–19.5) mg/dL; HDL-TG (10.3–15.4) mg/dL [66].

#### 4.3.4. Statistical Analyses

A descriptive analysis of the clinical and demographic variables and lipid profiles was carried out. Categorical variables were presented as the number of cases and percentages. Continuous variables were presented as mean and 95% confidence interval (95% CI). The normality of variables was assessed with graphs (QQ-Plot, density, and standard deviations), and only the distribution of TG was skewed and was log-transformed. A comparison of variables at baseline was performed using the Chi-square test for qualitative variables and analysis of variance for quantitative variables.

Differences in lipid and anthropometric variables between study groups were assessed over time using a linear mixed model. To correct for potential confounding, the study group’s comparison was adjusted by sex, age, lipid-lowering treatment, and smoking status. Moreover, an interaction between the study group and time was considered in order to model a differential evolution of the group effect over time. A post hoc analysis by sex was also performed by using linear mixed models adjusted by the same covariates except for sex. When appropriate, estimators are shown with 95% CI. All analyses were performed with a two-sided significance level of 0.05 and conducted with the use of R software version 3.6.1 to estimate the linear mixed model the package lmer4 was used [67].

## 5. Conclusions

In this sub-study of a lifestyle RCT, we compared the effects of MedDiet alone and an er-MedDiet+PA intervention on the lipoprotein subclass profiles of MetS patients. The MedDiet alone improved lipoprotein composition with a reduction in HDL-TG and sd-LDL-C and an increase in ILDL particles, while the er-MedDiet+PA intervention led to an improvement in TG metabolism, i.e., a decrease in VLDL-TG, VLDL-P, and lVLDL-P. It needs to be highlighted that these changes in TG metabolism are associated with an improvement in LDL composition. Thus, although no changes in LDL-P concentration were observed, VLDL-P with a higher content of TG decreased—a change that led to the formation of less atherogenic LDL-P. Further studies with longer follow-ups to evaluate the effect of energy-restricted diets on LDL-P concentration are warranted.

Both an energy-unrestricted MedDiet and er-MedDiet+PA intervention are promising strategies for reducing sd-LDL-C by optimizing different lipid metabolic pathways. These outcomes are important because sd-LDL-C is more atherogenic than lLDL-C. The combination of the MedDiet with a negative energy balance achieved by diet and physical activity has proven to be an effective approach for improving TG-rich lipoprotein metabolism and reducing the concentration of atherogenic lipoproteins. As the prescription of these diets must be conducted by experienced professionals and is time-consuming, further studies on their cost-effectiveness are warranted. Also, an assessment of the effects of the lipoprotein subclass changes induced by these diets on CVD and all-cause mortality is necessary.

Our results illustrate how lipid-lowering treatment alone without lifestyle changes is insufficient to reduce cardiovascular risk in patients with MetS.

## Figures and Tables

**Figure 1 ijms-25-01338-f001:**
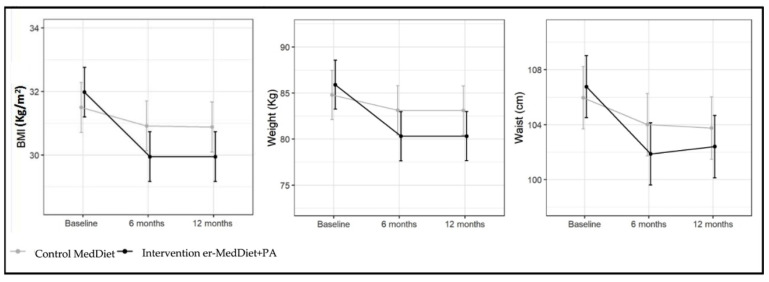
Effect of MedDiet and er-MedDiet+PA on anthropometric variables at 6 and 12 months. BMI (body mass index); MedDiet (Mediterranean diet); er-MedDiet +PA (energy-reduced MedDiet and physical activity).

**Figure 2 ijms-25-01338-f002:**
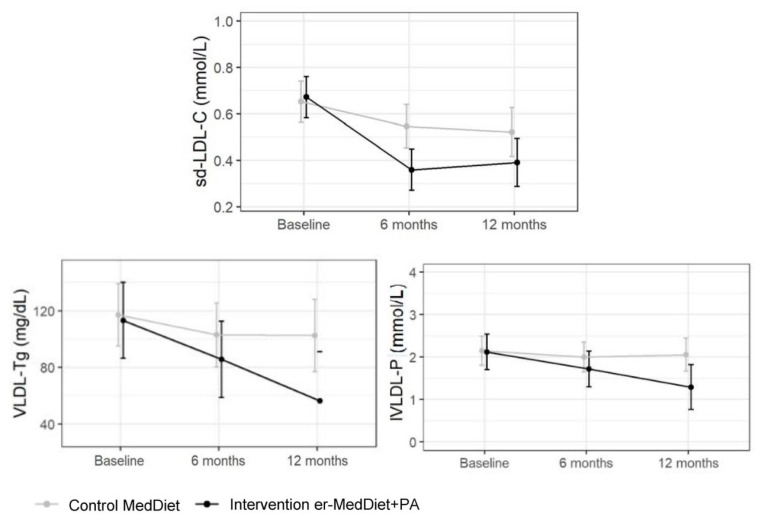
Effect of MedDiet and er-MedDiet+PA on sd-LDL-C, VLDL-TG, and lVLDL-P at 6 and 12 months. sd-LDL-C (small dense LDL Cholesterol); VLDL-TG (VLDL triglycerides); lVLDL-P (large VLDL particle number); MedDiet (Mediterranean diet); er-MedDiet +PA (energy-reduced MedDiet and physical activity). Data were analyzed by linear mixed models with intervention group, time (baseline, 6 and 12 months), interaction of group and time, and adjusted by sex, age, lipid-lowering treatment, and smoking status.

**Table 1 ijms-25-01338-t001:** Descriptive analysis of demographic, clinical, and anthropometric data.

	Control Group *n* = 95	Intervention Group *n* = 107	*p*-Value
Age (years)	64.2 (63.3, 65.2)	64.8 (63.9, 65.6)	0.427
Men (%)	49 (51.6%)	54 (50.5%)	0.875
Smoker (%)			0.774
Current smoker	12 (12.6%)	15 (14.0%)	
Former smoker	42 (44.2%)	42 (39.3%)	
Never smoker	41 (43.2%)	50 (46.7%)	
Education (%)			0.619
Elementary (5–8 years)	48 (50.5%)	61 (57%)	
Secondary (9–12 years)	26 (27.4%)	27 (25.2%)	
Post-secondary (>12 years)	21 (22.1%)	19 (17.8%)	
Hypertension (%)	79 (83.2%)	92 (86%)	0.578
Diabetes (%)	29 (30.5%)	37 (34.6%)	0.540
Dyslipidemia (%)	76 (80%)	82 (76.6%)	0.563
Lipid-lowering treatment (%)	55 (57.9%)	63 (58.9%)	0.887
Weight (kg)	85.4 (82.8, 87.9)	85.3 (82.9, 87.8)	0.997
Body mass index (kg/m^2^)	31.9 (31.4, 32.5)	32.0 (31.5, 32.6)	0.754
Waist circumference (cm)	107 (105.1, 108.8)	107 (105.3, 109)	0.925
Hip (cm)	107 (105.6, 108.6)	108 (106.2, 109.4)	0.555
Waist-to-hip ratio	1.00 (0.98, 1.02)	1.00 (0.98,1.01)	0.629
LDL-cholesterol (mmol/L)	3.04 (2.84, 3.23)	3.00 (2.83, 3.16)	0.765
LDL-cholesterol ≤ 3.37 mmol/L (%)	57 (66.3%)	71 (68.3%)	0.771
HDL-cholesterol (mmol/L)	1.24 (1.16, 1.31)	1.25 (1.18, 1.31)	0.802
HDL-cholesterol ≥ 1.04 mmol/L (%)	62 (65.3%)	75 (70.1%)	0.463
Triglycerides (mmol/L) *	1.73 (1.56, 1.92)	1.57 (1.43, 1.71)	0.152
Triglycerides ≤ 1.70 mmol/L (%)	48 (50.5%)	60 (56.1%)	0.430
Non-HDL-cholesterol (mmol/L)	3.81 (3.62, 4.0)	3.49 (3.25, 3.73)	0.045
Non-HDL-cholesterol ≤ 4.14 mmol/L (%)	63 (66.3%)	70 (68.6%)	0.729

LDL (low-density lipoprotein); HDL (high-density lipoprotein); non-HDL (non-high-density lipoprotein). Data are shown as a mean and 95% confidence interval or as absolute frequency and percentage and were analyzed by the chi-square test and analysis of variance, respectively. * Triglycerides have been analyzed by logarithmic transformation and are expressed as antilogarithms.

**Table 2 ijms-25-01338-t002:** Baseline energy, nutrient intake and physical activity, and changes at 6 and 12 months by treatment allocation.

Energy and Nutrients	Visit	Control Group *n* = 95	Intervention Group *n* = 107	*p*-Value
Energy–Kcal/day	Baseline	2352 (2213, 2490)	2343 (2221, 2464)	0.923
	6m.change	−215 (−338, −91)	−252 (−356, −148)	0.650
	12m.change	−250 (−378, −122)	−252 (−356, −148)	0.982
Protein—% Energy	Baseline	17.9 (17.1, 18.6)	17.6 (17.0, 18.3)	0.622
	6m.change	+0.7 (+0.03, +1.3)	+2.9 (+2.2, +3.5)	<0.001
	12m.change	+1.0 (+0.4, +1.7)	+2.7 (+2.0, +3.4)	0.001
Protein—g/kg of body weight	Baseline	1.21 (1.15, 1.27)	1.19 (1.14, 1.25)	0.682
	6m.change	−0.016 (−0.068, +0.036)	+0.17 (+0.116, +0.222)	<0.001
	12m.change	−0.02 (−0.083, +0.053)	+0.15 (+0.095, +0.204)	<0.001
Carbohydrate—% Energy	Baseline	37.7 (36.2, 39.2)	39.1 (37.8, 40.4)	0.165
	6m.change	−4.2 (−5.7, −2.7)	−6.6 (−7.9, −5.2)	0.021
	12m.change	−3.8 (−5.2, −2.3)	−7.0 (−8.4, −5.7)	0.001
Total fat—% Energy	Baseline	42.1 (40.8, 43.5)	40.4 (39.2, 41.6)	0.058
	6m.change	+3.6 (+2.1, +5.1)	+4.2 (+2.9, +5.5)	0.522
	12m.change	+2.5 (+1.1, +3.9)	+4.7 (+3.3, +6.2)	0.028
SFA—% Energy	Baseline	10.9 (10.4, 11.3)	10.2 (9.8, 10.6)	0.035
	6m.change	−0.7 (−1.2, −0.2)	−1.2 (−1.6, −0.8)	0.120
	12m.change	−1.3 (−1.8, −0.8)	−0.7 (−1.1,−0.3)	0.070
MUFA—% Energy	Baseline	21.8 (20.8, 22.8)	20.6 (19.8, 21.4)	0.057
	6m.change	+4.0 (+2.8,+5.1)	+5.8 (4.9, +6.8)	0.012
	12m.change	+3.3 (+2.2, +4.4)	+5.9 (+4.8, +6.9)	0.001
PUFA—% Energy	Baseline	7.2 (6.8, 7.6)	7.1 (6.7, 7.4)	0.638
	6m.change	+1.1 (+0.7, +1.5)	+1.4 (+1.0, +1.9)	0.204
	12m.change	+1.2 (0.7, +1.6)	+1.6 (+1.2, +2.0)	0.170
Fiber–g/day	Baseline	25.2 (23.7, 26.6)	24.5 (23.0, 26.1)	0.551
	6m.change	+2.7 (1.0, +4.5)	+5.7 (+3.8, +7.5)	0.026
	12m.change	+2.8 (+1.0, +4.3)	+4.9 (+3.2, +6.6)	0.067
Cholesterol–mg/day	Baseline	399 (375, 424)	396 (373, 419)	0.852
	6m.change	−25.6 (−49.0, −2.2)	−29.1 (−51.2,−6.9)	0.833
	12m.change	−27.8 (−52.5, −3.1)	−13.2 (−38.0, +11.6)	0.413
Alcohol intake–g/day	Baseline	8.1 (5.9, 10.3)	10.1 (7.6, 12.6)	0.243
	6m.change	−0.9 (−2.4, +0.5)	−2.2 (−3.9, −0.6)	0.251
	12m.change	+0.03 (−1.5, +1.6)	−1.9 (+3.8, −0.1)	0.115
Physical activity–MET min/week	Baseline	2336 (1957, 2715)	2614 (2240, 2987)	0.303
	6m.change	+404 (45, 763)	+1263 (847, 1681)	0.002
	12m.change	+569 (8, 1129)	+1242 (743, 1741)	0.076

Data are means (95% confidence intervals) and *p*-values by one-way analysis of variance. 6m.change (6 months change versus baseline); 12m.change (12 months change versus baseline); SFA (saturated fatty acids); MUFA (monounsaturated fatty acids); PUFA (polyunsaturated fatty acids); MET (metabolic equivalent).

**Table 3 ijms-25-01338-t003:** Effects of MedDiet and er-MedDiet+PA based intervention on lipid and anthropometric variables observed during follow-up visits.

	Time Effect (Control Effect *)				Interaction Group and Time (Intervention vs. Control Effect **)			
	6 Months		12 Months		6 Months		12 Months	
	Coefficient [95% CI]	*p*-Value	Coefficient [95% CI]	*p*-Value	Coefficient [95% CI]	*p*-Value	Coefficient [95% CI]	*p*-Value
Body mass index (kg/m^2^)	−0.58 [−0.84; −0.32]	<0.01	−0.62 [−0.88; −0.35]	<0.01	−1.45 [−1.82; −1.09]	<0.01	−1.42 [−1.78; −1.05]	<0.01
Waist circumference (cm)	−1.95 [−2.99; −0.91]	<0.01	−2.21 [−3.26; −1.15]	<0.01	−2.94 [−4.4; −1.48]	<0.01	−2.16 [−3.64; −0.68]	<0.01
Cholesterol								
LDL (mmol/L)	−0.1 [−0.26; 0.06]	0.212	−0.26 [−0.42; −0.1]	<0.01	0.04 [−0.18; 0.26]	0.727	0.2 3[0.01; 0.45]	0.044
sd-LDL (mmol/L)	−0.11 [−0.23; 0.02]	0.091	−0.13 [−0.26; 0]	0.049	−0.21 [−0.38; −0.04]	0.016	−0.15 [−0.33; 0.03]	0.108
HDL (mmol/L)	0.04 [0; 0.09]	0.047	0.02 [−0.03; 0.06]	0.446	0.04 [−0.02; 0.1]	0.233	0.1 [0.04; 0.17]	<0.01
Non-HDL (mmol/L)	−0.1 [−0.33; 0.13]	0.385	−0.37 [−0.62; −0.11]	<0.01	0.08 [−0.25; 0.4]	0.65	0.54 [0.17; 0.91]	<0.01
IDL (mg/dL)	−2.28 [−4.66; 0.09]	0.06	−2.72 [−5.43; −0.01]	**0.049**	−0.78 [−4.29; 2.72]	0.661	−1.97 [−6.43; 2.49]	0.386
VLDL (mg/dL)	−4.98 [−10.55; 0.6]	0.08	−4.11 [−10.51; 2.29]	0.208	−3.32 [−11.58; 4.94]	0.431	−10.38 [−21.02; 0.26]	0.056
VLDL+IDL (mg/dL)	−7.26 [−14.73; 0.21]	0.057	−6.87 [−15.44; 1.69]	0.116	−4.17 [−15.23; 6.89]	0.46	−12.19 [−26.38; 1.99]	0.092
Triglycerides								
LOG(TG (mmol/L))	−0.16 [−0.25; −0.08]	<0.01	−0.06 [−0.15; 0.03]	0.176	−0.06 [−0.18; 0.07]	0.385	−0.15 [−0.28; −0.02]	0.021
LDL (mg/dL)	−1.87 [−4.64; 0.9]	0.186	−2.45 [−5.6; 0.7]	0.127	−1.58 [−5.66; 2.51]	0.449	−2.7 [−7.86; 2.45]	0.304
HDL (mg/dL)	−4.19 [−8.2; −0.19]	0.04	−0.96 [−5.46; 3.54]	0.676	−1.48 [−7.32; 4.36]	0.62	−6.56 [−13.87; 0.74]	0.078
IDL (mg/dL)	−1.75 [−3.67; 0.17]	0.075	−1.8 [−3.99; 0.4]	0.108	−0.66 [−3.5; 2.17]	0.647	−2.23 [−5.84; 1.38]	0.226
VLDL (mg/dL)	−14.14 [−35.32; 7.04]	0.191	−14.5 [−38.78; 9.79]	0.242	−13.36 [−44.71; 17.99]	0.404	−42.33 [−82.56; −2.1]	0.039
Particle number								
LDL (nmol/L)	30 [−110.2; 170.21]	0.675	−111.06 [−269.85; 47.72]	0.17	−86.05 [−292.51; 120.41]	0.414	−41 [−299.64; 217.64]	0.756
lLDL (µmol/L)	17.31 [0.99; 33.63]	**0.038**	6.06 [−12.53; 24.66]	0.523	−3.26 [−27.35; 20.82]	0.791	−0.24 [−30.77; 30.29]	0.988
sLDL (nmol/L)	14.63 [−76.17; 105.43]	0.752	−87.05 [−189.62; 15.53]	0.096	−72.6 [−206.2; 61]	0.287	−40.29 [−206.9; 126.2]	0.635
HDL (µmol/L)	−2.67 [−5.3; −0.05]	0.046	−0.17 [−3.15; 2.8]	0.909	2.45 [−1.41; 6.31]	0.214	−0.81 [−5.66; 4.04]	0.744
lHDL (µmol/L)	−0.02 [−0.04; 0.01]	0.191	−0.01 [−0.04; 0.02]	0.489	0.02 [−0.02; 0.06]	0.3	−0.01 [−0.06; 0.03]	0.539
sHDL (µmol/L)	−1.7 [−3.81; 0.41]	0.115	0.12 [−2.28; 2.53]	0.92	1.06 [−2.05; 4.18]	0.503	−0.98 [−4.91; 2.96]	0.627
VLDL (nmol/L)	−12.06 [−26.75; 2.62]	0.107	−11.78 [−28.65; 5.09]	0.171	−3.18 [−25.27; 18.91]	0.778	−20.8 [−49.42; 7.82]	0.154
lVLDL (nmol/L)	−0.15 [−0.45; 0.16]	0.336	−0.1 [−0.45; 0.25]	0.588	−0.25 [−0.7; 0.2]	0.276	−0.73 [−1.32; −0.15]	0.014
sVLDL (nmol/L)	−10.87 [−25.99; 4.26]	0.159	−10.99 [−28.32; 6.34]	0.214	2.11 [−20.27; 24.49]	0.853	−10.02 [−38.7; 18.65]	0.493
Particle size								
VLDL (nm)	0.05 [−0.11; 0.2]	0.567	0.1 [−0.07; 0.28]	0.241	0.05 [−0.18; 0.28]	0.653	0.23 [−0.05; 0.51]	0.115
LDL (nm)	0.07 [−0.1; 0.24]	0.424	0.15 [−0.04; 0.34]	0.128	0.11 [−0.14;0.36]	0.376	0.05 [−0.26; 0.37]	0.74
HDL (nm)	0 [−0.03; 0.03]	0.904	−0.01 [−0.05; 0.02]	0.401	0 [−0.04; 0.04]	0.966	0.03 [−0.02; 0.08]	0.252

LDL (low-density lipoprotein); sd-LDL (small dense LDL); HDL (high-density lipoprotein); non-HDL (non-high-density lipoprotein); IDL (intermediate-density lipoprotein); VLDL (very low-density lipoprotein); VLDL+IDL (remnant lipoproteins); LOG(TG(mmol/L)) (logarithmic triglyceride transformation (mmol/L)); lLDL (large LDL); sLDL (small LDL); lHDL (large HDL); sHDL (small HDL); lVLDL (large VLDL); sVLDL (small VLDL); [IC95%] (95% confidence interval). Significant values are indicated with bold lettering. Data were analyzed by linear mixed models with intervention group, time (baseline, 6 and 12 months), interaction of group and time, and adjusted by sex, age, lipid-lowering treatment, and smoking status. * Time effect can be interpreted as the effect observed in the control group. ** Interaction group and time can be interpreted as the additional effect of the intervention group compared to the control group.

## Data Availability

The database is available upon request to the correspondence author.

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
