# Peer review of "The Impact of the Mediterranean Diet and Lifestyle Intervention on Lipoprotein Subclass Profiles among Metabolic Syndrome Patients: Findings of a Randomized Controlled Trial"

_ijms, 2024, doi:10.3390/ijms25021338_

Round 1
Reviewer 1 Report
Comments and Suggestions for Authors
Thanks for the opportunity to review this paper.
I have a couple of comments which I hope the authors will find useful.
Abstract:
line 49. Should specify that this is an additive decrease.
Is weight loss reduction more meaningful to report than BMI – noting objectives of the overall study related to weight loss?
“These objectives were achieving an average reduction of ≥8% of the initial body weight and an average reduction of ≥5% of initial waist circumference in the first 6-months, and maintaining these reductions throughout the duration of the study.”
Line 50. Abbreviate triglyceride consisted with other parameters that are abbreviated?
Results:
Line 111. How were the means of groups compared at baseline? Note – none of these statistical tests for comparisons at baseline are mentioned in the method
Line 114-120. Non-HDL was the only mean value that was significantly higher - need to clarify this in the text
Line 121-124 and Table 2. How compliant were participants with following a MedDiet - this is a main principal for comparison between groups especially noting there were similar levels of energy restriction seen in control and intervention (the intervention group was not meeting their target of 30% reduction) - much of the discussion talks about impact of the MedDiet but data are not shown.
Also what was their alcohol intake? Can impact lipids if intake is high even with a 12 hour fast.
Table 3. Makes the reader work to figure out what the total magnitude of change in the intervention group was! My preference would be to see data at each timepoint for both groups. Would also save repetition of information in figures (in main paper and supplementary files). And include weight data in the table.
Regarding figures – a little blurry and do not indicate any significance alluded to in text. If adjust table, they are not necessary and can be removed.
Line 163. Did participants meet % weight loss targets? Per objective
Discussion
Line 252-254 (and conclusion line 499-500): I’m not entirely sure how this can be concluded? Does “lipid-lowering treatment” refer to medication? I think this needs to be explored a little more as not all of your population were medicated (did adjust for this in model)? Or alternatively are you comparing to standard medication effects, and if so this should be mentioned in more detail and referenced?
Materials and Methods:
Line 355 consisted?
Line 358-9 suggest including objective target with this energy restriction (as above)
Line 379 minor grammar issue
Line 387 Mentions FFQ for diet data collection but not MedDiet scoring for adherence.
Line 426 reference for equation
Line 464-468 no mention of stats for baseline comparisons (Table 1) or those from table 2
Other comments:
Check consistency of abbreviations in text, figures and tables e.g have used sdLDL-C, sdLDL, sd-LDL-C
I have tended to see triglyceride abbreviated to TG or TAG, not often Tg?
Author Response
Reviewer 1
I have a couple of comments which I hope the authors will find useful.
Abstract:
line 49. Should specify that this is an additive decrease.
Is weight loss reduction more meaningful to report than BMI – noting objectives of the overall study related to weight loss?
RESPONSE: Following your helpful suggestion, the sentence in line 49: “Compared to the control diet, at 12 months the er-MedDiet+PA resulted in a significant decrease of body mass index by 1.4 kg/m2, reduction of waist circumference…”
Has been changed to: “Compared to the control diet, at 12 months the er-MedDiet+PA resulted in a significant additional 4,2 kg of body weight loss, a decrease of body mass index by 1.4 kg/m2, reduction of waist circumference by 2.2 cm…”
“These objectives were achieving an average reduction of ≥8% of the initial body weight and an average reduction of ≥5% of initial waist circumference in the first 6-months, and maintaining these reductions throughout the duration of the study.”
RESPONSE: Thank you for this kind indication. According to it, in the Study design section, the sentence: “In summary, the intensive weight loss program consists of an er-MedDiet together with the promotion of physical activity and behavioral support for specific weight loss goals.” has been changed to: “In summary, the intensive weight loss program consisted of an er-MedDiet together with the promotion of physical activity and behavioral support for specific weight loss goals that included an average reduction in baseline body weight of over 8% and an average reduction in waist circumference of over 5% in the first six months and maintaining these reductions throughout the duration of the study.”
Line 50. Abbreviate triglyceride consisted with other parameters that are abbreviated?
RESPONSE: Taking into account your question, the abbreviation "Tg" has been changed throughout the article to the abbreviation "TG"
Results:
Line 111. How were the means of groups compared at baseline? Note – none of these statistical tests for comparisons at baseline are mentioned in the method
RESPONSE: According to your helpful comment, we have added in the section “Statistical analysis” the sentence: “A comparison of variables at baseline was performed using the Chi-square test for qualitative variables and analysis of variance for quantitative variables.”
Line 114-120. Non-HDL was the only mean value that was significantly higher - need to clarify this in the text
RESPONSE: Thank you for this comment. The sentence: “However, mean LDL-C and mean HDL-C were similar between groups and mean TG concentration were slightly higher in the intervention group.” has been changed to: “However, mean LDL-C and mean HDL-C were similar between groups, and mean non-HDL-C was higher in the control group (p= 0.045).”
Line 121-124 and Table 2. How compliant were participants with following a MedDiet - this is a main principal for comparison between groups especially noting there were similar levels of energy restriction seen in control and intervention (the intervention group was not meeting their target of 30% reduction) - much of the discussion talks about impact of the MedDiet but data are not shown.
RESPONSE: Thank you for this kind observation. The following sentence has been added in the Discussion (Anthropometric variables) section: “Although patients in the intervention group did not reduce total energy intake, they increased energy expenditure and improved diet quality, with a marked decrease in carbohydrate intake and an increase in monounsaturated fat consumption“.
Also what was their alcohol intake? Can impact lipids if intake is high even with a 12 hour fast.
RESPONSE: Thank you for your question. Data on mean alcohol intake showed that it was not high in both the intervention and control groups. We have added data on alcohol consumption in table 2.
Table 3. Makes the reader work to figure out what the total magnitude of change in the intervention group was! My preference would be to see data at each timepoint for both groups. Would also save repetition of information in figures (in main paper and supplementary files). And include weight data in the table.
Regarding figures – a little blurry and do not indicate any significance alluded to in text. If adjust table, they are not necessary and can be removed.
RESPONSE: We appreciate your kind and helpful comments. We would like to highlight that the objective of the statistical analysis was to evaluate the effect of interventions on lipoprotein subclass profiles using multivariate mixed models. To quantify this effect and determine whether it was significant, we adjusted the regression models for sex, age, lipid-lowering treatment, and smoking. According to this type of analysis, it seems better to express the results in terms of model coefficients, since the means may not be a faithful reflection of the model used in the statistical study carried out. We would also like to maintain the figures to facilitate the reader's understanding of the effect of both interventions.
Line 163. Did participants meet % weight loss targets? Per objective
RESPONSE: According to your kind question, the following information has been added in the Results (Anthropometric variables) section: “At 6 months, 4.2% of subjects in the control group and 38.5% of subjects in the intervention group achieved weight loss of at least 8%, and these percentages were 5.3% and 39.8%, respectively, at 12 months.”
Discussion
Line 252-254 (and conclusion line 499-500): I’m not entirely sure how this can be concluded? Does “lipid-lowering treatment” refer to medication? I think this needs to be explored a little more as not all of your population were medicated (did adjust for this in model)? Or alternatively are you comparing to standard medication effects, and if so this should be mentioned in more detail and referenced?
RESPONSE: We appreciate your kind question. As indicated in the methods section (statistical analysis) and at the bottom of the respective tables, the model was adjusted for different variables, including lipid-lowering drug therapy.
Materials and Methods:
Line 355 consisted?
RESPONSE: Thank you for this observation. Accordingly, the word “consists” has been changed to “consisted”.
Line 358-9 suggest including objective target with this energy restriction (as above)
RESPONSE: We appreciate your suggestion. Details on limitations to the consumption of these foods are described in the project protocol mentioned above on the same page.
Line 379 minor grammar issue
RESPONSE: According to your kind observation, “Only those visited at the Hospital Universitari de Bellvitge…” has been changed to “Only those participants visited at the Hospital Universitari de Bellvitge…”.
Line 387 Mentions FFQ for diet data collection but not MedDiet scoring for adherence.
RESPONSE: Following your helpful suggestion, the following sentence has been added in the Methods section: “The adherence to MedDiet was assessed by a 17-item questionnaire [65]”.
Line 426 reference for equation
RESPONSE: Thank you for this observation. The following reference has been added: [69]. Friedewald WT, Levy RI, Fredrickson DS. Estimation of the concentration of low-density lipoprotein cholesterol in plasma, without use of the preparative ultracentrifuge. Clin Chem. 1972, 18;499-502.
Line 464-468 no mention of stats for baseline comparisons (Table 1) or those from table 2
RESPONSE: We appreciate your helpful comment. We have added the following sentence in the Statistical analysis section: “A comparison of variables at baseline was performed using the Chi-square test for qualitative variables and analysis of variance for quantitative variables.”
Other comments:
Check consistency of abbreviations in text, figures and tables e.g have used sdLDL-C, sdLDL, sd-LDL-C
RESPONSE: Following your kind directions, the abbreviation “sdLDL” has been changed throughout the article to the abbreviation “sd-LDL”.
I have tended to see triglyceride abbreviated to TG or TAG, not often Tg?
RESPONSE: Thank you for this comment. The abbreviation "Tg" has been changed throughout the article to the abbreviation "TG".

Reviewer 2 Report
Comments and Suggestions for Authors
I congratulate the authors for this very innovative research project in the Lipidology Science field.
Here are my suggestions to improve the quality of your work:
1. Table 2
Is it possible to add a row for proteins?
It would be very interesting to observe a Protein (g/kg of body weight) for the intervention vs control group at the 3 times.
The second question is: " Why do the authors not insert a row with Carbohydrates % Energy?
With a brief calculation in the intervention group (post 12 months) I note that the carbohydrate content is near 35% of total energy. This is due to considering that this intervention diet is not only a Mediterranean diet, but even a Low-carb Mediterranean Diet.
My suggestion is to report this consideration in the Title and the other relevant parts of your manuscript.
2. Page 5 is empty, please consider deleting it.
3. Table 3, The authors should explain better what is intended for Triglycerides, LOG(Tg(mmol/L))? The explanation should be inserted in the text that comments the Table 3 under the description of the table with the abbreviations.
4. Line 405, please specify the name of the detergent used.
Comments on the Quality of English Language
NONE
Author Response
Review 2
I congratulate the authors for this very innovative research project in the Lipidology Science field.
Here are my suggestions to improve the quality of your work:
- Table 2
Is it possible to add a row for proteins? It would be very interesting to observe a Protein (g/kg of body weight) for the intervention vs control group at the 3 times.
RESPONSE: Following your kind suggestion, the amount of protein (g/kg of body weight) consumed by the study subjects has been added in table 2.
The second question is: " Why do the authors not insert a row with Carbohydrates % Energy?
RESPONSE: Thank you for your kind question. The information on Carbohydrates % energy is included in table 2.
With a brief calculation in the intervention group (post 12 months) I note that the carbohydrate content is near 35% of total energy. This is due to considering that this intervention diet is not only a Mediterranean diet, but even a Low-carb Mediterranean Diet. My suggestion is to report this consideration in the Title and the other relevant parts of your manuscript.
RESPONSE: Thank you for your very kind and interesting comment. As it is stated in the Predimed-Plus research plan (https://www.predimedplus.com/wp-content/uploads/2018/11/Protocolo-PREDIMED-Plus_Eng.pdf), the diet used in the Predimed study and in the Predimed-Plus study was considered a traditional Mediterranean diet (Estruch R, Martínez-González MA, Corella D, et al. Effect of a high-fat Mediterranean diet on bodyweight and waist circumference: a prespecified secondary outcomes analysis of the PREDIMED randomised controlled trial. Lancet Diabetes Endocrinol 2016;4:666-76.)(Estruch R, Ros E, Salas-Salvado J, et al. Primary prevention of cardiovascular disease with Mediterranean diets: the PREDIMED trial. 2013;368:1279-90). Mediterranean diet is rich in fat from vegetable sources (virgin olive oil and nuts) that can be more useful for developing and implementing programs aimed at achieving prolonged weight loss and improving the metabolic alterations associated with overweight and obesity. In contrast, other studies conducted with low-fat diets, such as the Look Ahead trial, that addressed the long-term effect of an intensive weight-loss lifestyle program in obese adults, that included a low-fat diet (<30% of total energy intake with <10% from saturated fat), on CVD and mortality ended prematurely due to lack of efficiency (Look AHEAD Research Group. Cardiovascular effects of intensive lifestyle intervention in type 2 diabetes. N Engl J Med 2013;369:145-54). The low-fat diet in the Women’s Health Initiative Dietary Modification Trial (Howard BV, Van Horn L, Hsia J, et al. Low-fat dietary pattern and risk of cardiovascular disease: the Women's Health Initiative Randomized Controlled Dietary Modification Trial. JAMA 2006;295:655-66.) also did not demonstrate beneficial effects. These studies used diets somewhat opposite to the traditional Mediterranean diet used in the PREDIMED and in the Predimed-Plus trials which contained a similar proportion of fat, close to 40% (predominantly unsaturated), and carbohydrates.
- Page 5 is empty, please consider deleting it.
RESPONSE: Done.
- Table 3, The authors should explain better what is intended for Triglycerides, LOG(Tg(mmol/L))? The explanation should be inserted in the text that comments the Table 3 under the description of the table with the abbreviations.
RESPONSE: Following your helpful directions, the description of LOG(Tg(mmol/L)) has been added at the bottom of the table with the abbreviations.
- Line 405, please specify the name of the detergent used.
RESPONSE: Thank you for this kind indication. Unfortunately, we do not know the specific detergent used as it is part of Roche Diagnostics patented reagent and the manufacturer does not specify it.

Round 2
Reviewer 2 Report
Comments and Suggestions for Authors
The authors reviewed the paper following the suggestion of reviewers.
Comments on the Quality of English LanguageOk.